# Path Planning and Formation Control for UAV-Enabled Mobile Edge Computing Network

**DOI:** 10.3390/s22197243

**Published:** 2022-09-24

**Authors:** Kheireddine Choutri, Mohand Lagha, Souham Meshoul, Samiha Fadloun

**Affiliations:** 1Aeronautical Sciences Laboratory, Aeronautical and Spatial Studies Institute, Blida 1 University, Blida 0900, Algeria; 2Department of Information Technology, College of Computer and Information Sciences, Princess Nourah bint Abdulrahman University, P.O. Box 84428, Riyadh 11671, Saudi Arabia; 3Ecole Nationale Supérieure d’Informatique (ESI), Alger 16309, Algeria

**Keywords:** Mobile Edge Computing, UAV, formation control, leader election, path planning, obstacle avoidance, artificial intelligence, multi-agent systems

## Abstract

Recent developments in unmanned aerial vehicles (UAVs) have led to the introduction of a wide variety of innovative applications, especially in the Mobile Edge Computing (MEC) field. UAV swarms are suggested as a promising solution to cope with the issues that may arise when connecting Internet of Things (IoT) applications to a fog platform. We are interested in a crucial aspect of designing a swarm of UAVs in this work, which is the coordination of swarm agents in complicated and unknown environments. Centralized leader–follower formations are one of the most prevalent architectural designs in the literature. In the event of a failed leader, however, the entire mission is canceled. This paper proposes a framework to enable the use of UAVs under different MEC architectures, overcomes the drawbacks of centralized architectures, and improves their overall performance. The most significant contribution of this research is the combination of distributed formation control, online leader election, and collaborative obstacle avoidance. For the initial phase, the optimal path between departure and arrival points is generated, avoiding obstacles and agent collisions. Next, a quaternion-based sliding mode controller is designed for formation control and trajectory tracking. Moreover, in the event of a failed leader, the leader election phase allows agents to select the most qualified leader for the formation. Multiple possible scenarios simulating real-time applications are used to evaluate the framework. The obtained results demonstrate the capability of UAVs to adapt to different MEC architectures under different constraints. Lastly, a comparison is made with existing structures to demonstrate the effectiveness, safety, and durability of the designed framework.

## 1. Introduction

The majority of Internet of Things (IoT) applications require a number of computational tasks that cannot be performed locally on IoT devices due to the vast amount of data that are generated and need to be processed in real time. Because of their greater processing and storage capacity, external devices in centralized servers in remote clouds or at the edge are used to perform these IoT tasks, which require offloading operations. Connecting an Internet of Things network to a Mobile Edge Computing (MEC) platform, on the other hand, could be a challenging endeavor. The use of unmanned aerial vehicles (UAVs) is one possible solution to this issue. A survey concerned UAV-Enabled Mobile Edge Computing for IoT Devices can be found on [1]. Depending on their role in the network, UAVs can be used as fog nodes [2] or as relays to improve the connectivity of a wireless network [3]. In [4], a swarm of UAVs is employed in the fog layer to detect objects in real time videos.

UAVs with onboard computing resources can provide offloading services to nearby mobile users (MUs). Within the constraints of the available computing power, numerous strategies were proposed to reduce the processing time among MUs. While [5] offers a compute offloading strategy based on Deep Deterministic Policy Gradient (DDPG), Ref. [6] uses Reinforcement Learning (RL) to enhance the Quality-of-Service for each terminal user. Depending on the size and weight of the UAV, work offloading in UAV-enhanced edges can be challenging. Due to the influence that UAV-reliant networks may have on QoS and performance, it is vital that they be designed with energy efficiency in mind. The authors of [7] offer an approach that simultaneously optimizes the offloading of tasks and the UAV’s flying path. Two objective functions addressing energy-efficient offloading and safe path planning are proposed in [8] as part of a constrained multi-objective optimization problem for UAVs. The authors of [9] are primarily concerned with path planning under the paradigm of edge computing. In addition, the authors of [10] present an algorithm for creating a feasible task migration path, taking into account the migration distance between UAVs, the load condition of UAVs, and environmental characteristics. The authors of [11] describe a cloud-based motion planning method for coordinating the safe movement of a large number of mobile robots.

The mobility of UAVs and their ever-increasing computational power paved the path for their usage in a wide variety of novel applications. Recently, a significant amount of work and attention has been focused on their application in the support of mobile edge computing systems. Several different MEC systems that are aided by UAVs have been proposed [12]. On the other hand, it has been reported that a single UAV-based solution might not be adequate to complete the mission due to the limited life of its battery and the limited computing power it possesses. A swarm of drones has many advantages over a single drone, including increased reliability and decreased operation time. As a consequence, there has been a surge in the use of swarms of drones rather than a single drone. Dealing with a swarm of drones gives rise to a number of challenges that need to be addressed among which, are multi-agent systems, path planning, and formation control.

Multi-agent systems have received much attention from the scientific community in recent years. Multiple collaborating UAVs can complete more challenging jobs and achieve more complicated objectives than a single UAV. Various techniques and designs have been proposed in the literature, including behavior-based [13], virtual structure [14], potential field [15], and leader–follower [16,17,18]. In the centralized leader–follower (L-F) scenario, a “leader” agent has the reference motion that the other agents, “followers”, track it.  To facilitate collaboration, the leader conveys its state to the followers via an appropriate communication channel or link. Consequently, a leader’s failure will result in mission failure. The leader election is one of the recently researched challenges in leader–follower settings. In most cases, the leader is assumed to be a specific agent picked at the start of the job; this event is known as a static leader election. Ref. [19] offers a new leader election approach based on an adaptive/reliable network structure. Ref.  [20] offers a distributed leader election method that does not require direct inter-agent communication. For online leader election, the authors of [21] suggest a fully-decentralized adaptive technique capable of selecting the best leader among the existing leader’s neighbors on a regular basis. This author describes a distributed online leader election model. The swarm agents are adaptive to unexpected leader failure and changes in network topology.

In formation control, the motion of UAVs is tightly restricted to preserve the formation topology. Consensus algorithms have been explored extensively in the literature for this purpose [22,23]. Numerous control mechanisms were proposed for UAV formation control, with the underlying theory described in [24]. Ref. [25] suggests a second-order consensus algorithm to follow a specified external reference in their work, whereas [26] frames the formation control problem as a position control problem to be solved. In contrast, spacecraft formation vehicles use a mechanism for robust attitude control, as described in [23]. The prior control strategies were able to sustain the formation, requiring a position estimate for both the leader and the followers. In addition, communication limitations pertaining to the proposed typologies were not considered. A consensus-based attitude formation controller is utilized for this study. The formation topology is then preserved with minimal data interchange, and the controller is resilient to external disturbances.

Avoiding other UAVs and natural obstacles is essential for mission success and safe operation. There have been numerous proposed solutions to this problem. The avoidance of obstacles by unmanned aerial vehicles is examined in [27]. In [28], the methods of conflict identification and resolution for a multi-UAV collision-avoidance system are investigated. Therefore, a modified technique is given for tentacle construction and collision avoidance for multiple UAVs operating in unstructured environments. For use in the actual world without the assistance of a pilot, Ref. [29] developed a system that can automatically navigate and avoid obstacles. Moreover, Ref. [30] describes a decentralized control system based on behavior, which employs a sliding mode controller and artificial potential functions. Although other works have addressed the issue of avoiding obstacles when flying in a UAV formation, nobody has yet optimized the resulting trajectory.

This research offers a new paradigm for multi UAVs, one that makes distributed formation control operable across a wide range of MEC architectures, unlike previous work in [31,32,33,34]. Therefore, algorithms for optimal trajectory tracking, obstacle avoidance, and leader selection were incorporated into the resulting framework. The suggested framework removes the problems associated with conventional leader–follower development and produces better results. The following are the foundational requirements of the method we propose:**R1**: ***Multi-agent systems (MAS):*** Multi-agent systems are computational systems composed of a large number of interacting computer elements known as agents. To take advantage of the decentralized structure of the multi-agent system, the agent must be provided with some autonomy. When we state that an agent is autonomous, we infer that it can collect data by interacting with other agents and its environment, and then make decisions based on this information.**R2**: ***Formation control***: According to the proposed method, swarms have a certain topology, such as a rectangle, a diamond, etc. This allows the formation topology to be maintained with little data sharing and makes the controller more resilient to external disturbances.**R3**: ***Leader election***: In centralized formations, there is often one leader of the swarm; nevertheless, if a single error happens, the entire mission is aborted. In this situation, the other agents must select a new leader.**R4**: ***Obstacle avoidance***: In centralized formations, there is often one leader of the swarm; nevertheless, if a single error happens, the entire mission is aborted. In this situation, the other agents must select a new leader.

The remainder is organized as follows: In Section 2, the system is introduced with a brief overview of graph theory and MEC architectures. The problem of path planning and obstacle avoidance is formulated in Section 3. Section 4 is dedicated to formation control. The formalization of the dynamic model using quaternions is described first. Then, a suitable SMC controller is developed to preserve the topology of the formation. Finally, the formation transformation for leader election is described. Section 5 examines the simulation results, proposing numerous scenarios and comparing performance. Section 6 presents conclusions and future potential work.

## 2. Preliminaries

### 2.1. Mec Architectures

Depending on the role of UAVs in the network, three distinct MEC topologies can be distinguished. In each scenario, the MEC server and MUs are the primary components, which are modeled as data analysis centers located closer to the users on the network’s edge. Using wireless networking and accessible communication technologies, links are built between the different users and servers. Internet is then used to link the MEC server to cloud-based data analysis centers [1].

***Assisted MEC:*** This architecture is often used to provide services following a natural disaster or bombardment-related infrastructure damage. As depicted in Figure 1, the UAV assists mobile users by acting as an aerial MEC server-enabled base station. Each user transfers its computationally intensive activities to one or many UAVs for processing. Therefore, UAVs with long-lasting batteries and powerful CPUs are necessary for this architecture. In addition, this architecture is typically employed to satisfy QoS requirements by optimizing the overall energy consumed by the MUs.***Cellular-Connected MEC:***Figure 2 illustrates these kind of architectures. During a mission, UAVs are viewed as aerial users with computationally intensive tasks, such as path planning and data analysis. Due to limited onboard processing capability, UAVs offload computation to an MEC server on a Ground Base Station (GBSs). In comparison to the previous architecture, the UAVs deployed in this manner have limited batteries and possessors, but they must conduct intense computation tasks.***Relayed MEC:*** As shown in Figure 3, the UAVs in this final architecture serve as relays to help the MUs in offloading their intensive computation tasks to the MEC server of the GBSs. Therefore, none of the UAVs include an MEC server. This architecture is intended to enable long-distance communication links between the MUs and the MEC server in the event that other regular links are interrupted.

### 2.2. System Presentation

The proposed framework consists of a swarm of UAVs assisting different MEC architectures in a distributed formation. We assume that only the leader is aware of the mission task and all possible formation types in the L-F approach. Followers are expected to collect data from other UAVs and the surrounding environment. The leader takes decisions on specific movements, planned trajectory, and allocation of tasks. The framework is constructed according to the flow chart shown in Figure 4.

The mission is first planned by the user, who specifies the departure and arrival points, the number of UAVs, the formation topology, and the employed MEC architecture. The path planning stage then generates the best path between the departure and arrival points. The control space is optimized, while some constraints in the output and state spaces are applied. This includes physical constraints, actuator limitations, and environment obstacles. Next, the formation control phase aims to preserve the formation’s topology and accommodate any topological variations caused by constraints. Both the leader and the followers have an autopilot system for attitude stabilization and trajectory tracking. As a result, the leader receives the required position to control command for the autopilot in order to arrive at the desired location. The main objective of the UAVs is to offload their computation tasks to/from the MEC server depending on the selected architecture. However, if a failure happens during the mission, the framework includes a safety stage to cope with it. If the leader is unable to complete the task, the followers elect a new leader. The technique seeks to select the most qualified UAV for leadership and switch topologies based on the number of remaining UAVs. Furthermore, if an external obstacle is detected, the leader instructs the followers to split the topology, avoid the obstacle, and return to the original topology.

In this study, we represent the network topology of the UAV swarm using graph theory notations. Nodes in an undirected graph represent network communication nodes, while edges represent communication links between nodes. In the case of UAV swarms, nodes represent individual UAVs and edges represent inter-UAV links, such as wireless communications or sensors. We assume that all communication links are bidirectional.

### 2.3. Graph Theory

Abstractly, an undirected graph G=(ν,ε) is defined by a node set *v* with cardinality *n*, the number of nodes in the graph, and an edge set E comprised of pairs of nodes, where nodes $\nu_{i}$ and $\nu_{j}$ are adjacent if $\left\{\nu_{i},\nu_{j}\right\}\in \varepsilon \subseteq {\left[v\right]}^{2}$. One special family of undirected graphs are tree graphs, denoted by the set Γ, where all two-node pairs are connected by exactly one simple path, that is, a connected graph without cycles. A spanning tree of a graph is a tree sub graph that connects all vertices in the graph.

The neighborhood set $N(v_{i})$ of a node $\nu_{i}$ is composed of the set of nodes adjacent to $\nu_{i}$. The scalar $d=(\nu_{i},\nu_{j})$ is the minimum path length, induced by the graph *G*, between nodes $\nu_{i}$ and $\nu_{j}$. The adjacency matrix of *G* is a positive matrix denoted by $G^{A}=\left[\omega_{ij}^{a}\right]\in R^{nn}$, where $\omega_{ij}^{a}$ represents the entry of the *i*-th row and *j*-th column of matrix $G^{A}$ with $\omega_{ij}^{a}=1$ if (i,j)∈E, and $\omega_{ij}^{a}=0$, otherwise. The degree $\delta_{i}$ of node $\nu_{i}$ is the number of adjacent nodes. The degree matrix $G^{D}$ is a diagonal matrix with $\delta_{i}$ at matrix element (i,i). We also define the diagonal matrix $G^{L}=diag\{\omega_{1}^{l},\ldots ,\omega_{n}^{l}\}$ representing the status of the agent—if $\omega_{i}^{l}=1$, then agent *i* is a leader. Otherwise, if $\omega_{i}^{l}=0$, then agent *i* is a follower.

The Laplacian matrix graph is defined as $L=G^{D}-G^{A}\in R^{nn}$. It plays an important role in the dynamics of the network. An important feature of this matrix is that it is a (symmetric) positive semi-definite matrix. The spectrum is ordered as $0=\lambda_{1}\left(L\right)\leq \lambda_{2}\left(L\right)\leq \ldots \leq \lambda_{n}\left(L\right)$. The interaction positive matrix *G* for the L-F formation is defined as follows:(1)$$G=G^{D}-G^{A}+G^{L}=L+G^{L}$$

Based on graph theory, the L-F consensus’ desired trajectory for each agent can be given by:(2)$$\begin{array}{c} \begin{array}{ccc} {\bar{x}}_{i}^{d}=\frac{1}{|N_{i}|}\sum_{j\in N_{i}}(\left(x_{j}+d_{ij}\right) & \; & \;i\;\mathrm{follower}\\ {\bar{x}}_{i}^{d}=\frac{1}{|N_{i}+1|}(\sum_{j\in N_{i}}\left(\left(x_{j}+d_{ij}\right)+r+d_{i0}\right) & \; & \;i\;\mathrm{leader} \end{array} \end{array}$$
where $d_{ij}=d_{i0}-d_{j0}$ is the inter-distance. Once an observation is available, Equation (Equation 2) is reformulated based on the normalized interaction matrix $\tilde{G}$ as follows:(3)$$\begin{array}{c} \left[\begin{array}{c} x_{i}-{\bar{x}}_{i}^{d}\\ \vdots \\ x_{n}-{\bar{x}}_{n}^{d} \end{array}\right]=\left(\tilde{G}⊗I_{2}\right)\left[\begin{array}{c} x_{i}-x_{i}^{d}\\ \vdots \\ x_{n}-x_{n}^{d} \end{array}\right] \end{array}$$

For this study, a normalized interaction matrix is considered for L-F formations as indicated in Equation (Equation 4):(4)$$\begin{array}{c} \tilde{G}={\left(G^{D}+G^{L}\right)}^{-1}\cdot G\; \end{array}$$

Extra edges can be added to the proposed topology as shown in Figure 5. The added two edges and the same vertices constitute the graph $G_{e}=\left(\nu ,{\left(\epsilon \right)}_{e}\right)$. Then, $G_{e}$ and $\tilde{G}$ yield:(5)$$\begin{array}{c} \tilde{G}=G+G_{e} \end{array}$$

## 3. Path Planning Stage

The main objective of the path planning stage is to generate the optimal path between the departure and arrival points. A collision avoidance system should often be incorporated into the control architecture to protect UAVs from causing damage to themselves and surrounding objects (other vehicles, persons, and infrastructures). Thus, obstacle avoidance techniques are used to tackle problems with trajectory optimization challenges. The majority of issues in the literature are addressed using dynamic programming, with trajectory optimization occurring throughout the challenge. In this study, however, the obstacle avoidance problem is seen as a Constrained Optimization (CM) problem, with the objective of optimizing the problem’s key parameters: the UAVs desired inter-distance ($d_{d}$) and the radius of the obstacle avoidance zone ($R_{max}$).

Next, an energy-aware communication and computation resource is incorporated for the aim of minimizing the total energy consumption. First, the energy consumed by MUs and UAVs while processing tasks is included in the task processing energy consumption. According to capacitance theory [10], the processing energy consumption of an MU or UAV is mostly governed by the CPU performance of the electronic device. Secondly, the communication energy consumption is made up of the energy consumed by the MU’s offload task, as well as the energy consumed by switching topologies between UAVs. Finally, the problem of minimizing the energy consumption by engines and electronic devices is also considered. To summarize, the total energy consumption of all UAVs in the system during a time step *t* can be defined by Equation (Equation 6):(6)$$\begin{array}{c} E_{UAV}^{t}=\sum_{n\in N}E_{Com,n}^{t}+E_{Tran,n}^{t}+E_{Fly,n}^{t} \end{array}$$
where:

$E_{Com}^{t}$ is the energy consumption for processing offloaded tasks of UAV *n*;

$E_{Tran}^{t}$ is the transmission energy consumption of MU m offload tasks to UAV *n*;

$E_{Fly}^{t}$ is the flight energy consumption of UAV *n*.

The optimization procedure is guided by an aggregation of two objective functions, namely the minimal distance traveled ($e_{p}$) and the minimum energy (*E*). Consequently, the objective of this work is to minimize traveled distance and energy consumption, while avoiding obstacles and collisions. This can be formulated as follows:(7)$$\begin{array}{c} \begin{array}{cc} min & \Phi =\int_{0}^{T}\left(e_{p}^{T}\cdot W\cdot e_{p}+E^{T}\cdot Q\cdot E\right)dt\\ s.t. & \\ \; & E\leq E_{max}\\ \; & R\geq R_{max}\\ \; & d=d_{d} \end{array} \end{array}$$
where *W* and *Q* are the weighting matrix. Traditional methods can be used to solve the problem. Karush–Kuhn–Tucker conditions are utilized because of inequality  limitations.

## 4. Formation Control Stage

### 4.1. UAV Model

Quadrotors are a subset of Vertical Take-off and Landing (VTOL) UAVs. Their ability to hover, move forward, and perform vertical takeoffs and landings make them nonlinear and under-actuated systems. This work supposes an attitude representation under quaternions approach to avoid singularities presented in Euler angle presentation. Let the unit quaternion $q\left(t\right)\in H,\bar{q}\left(t\right)\in R^{3},q_{0}\left(t\right)\in R$ be defined with:(8)$$\begin{array}{c} q\left(t\right)=q_{0}\left(t\right)+\bar{q}\left(t\right)=q_{0}\left(t\right)+{\left[q_{1}\left(t\right),q_{2}\left(t\right),q_{3}\left(t\right)\right]}^{T} \end{array}$$
with $q_{0}\left(t\right)$ a given quaternion and $\bar{q}\left(t\right)$ the complex and $q_{0}\left(t\right)$ the scalar parts of q(t). The unit quaternion should satisfy:(9)$$q_{0}{\left(t\right)}^{2}+∥\bar{q}\left(t\right)∥=1$$

Finally, the quadrotors dynamic model is:(10)$$\ddot{p}=q⊗\frac{F}{m}⊗q^{*}+\bar{g}$$
(11)$$\dot{q}=\frac{1}{2}q⊗w$$
(12)$${\dot{\omega }}_{b}=J^{-1}\left(\tau -\omega_{b}*J\omega_{b}\right)$$

The quaternion derivative is derived based on the angular velocity ω(t). The designed control inputs are given by Equation (Equation 13) with $k_{T}$ as the thrust constant, $k_{D}$ the the drag constant, and *l* the distance from the motor axis to the quadrotors center of mass:(13)$$\left[\begin{array}{c} U_{1}\\ U_{2}\\ U_{3}\\ U_{4} \end{array}\right]=\left[\begin{array}{c} k_{T}\left(\omega_{1}^{2}+\omega_{2}^{2}+\omega_{3}^{2}+\omega_{4}^{2}\right)\\ lk_{T}\left(\omega_{1}^{2}-\omega_{2}^{2}-\omega_{3}^{2}+\omega_{4}^{2}\right)\\ lk_{T}\left(\omega_{1}^{2}+\omega_{2}^{2}-\omega_{3}^{2}-\omega_{4}^{2}\right)\\ k_{D}\left(-\omega_{1}^{2}+\omega_{2}^{2}-\omega_{3}^{2}+\omega_{4}^{2}\right) \end{array}\right]$$

Based on Lyapunov stability analysis, the position and velocity control laws can be derived as indicated in [26] as follows:(14)$$\begin{array}{c} v_{d}=c_{1P}e_{1}+{\dot{p}}_{d} \end{array}$$
(15)$$\begin{array}{c} \ddot{p}=e_{1}\left(I-c_{1P}^{2}\right)+e_{2}\left(c_{1P}+c_{2P}\right)+{\ddot{p}}_{d} \end{array}$$
where $e_{1}=p_{d}-p$ is the position error, $e_{2}=v_{d}-v$ is the velocity error, and $c_{1P}$, $c_{2P}$ are positive constant coefficients.

### 4.2. Formation Control

After following the leader for the same or different altitude, the formation controller’s objective is to maintain the X-Y topology. This is accomplished by maintaining a specific distance *d* and angle α between the leader and each follower. Thus:(16)$$\begin{array}{c} d_{x}=-\left(X_{L}-X_{F}\right)\cos\psi_{L}-\left(Y_{L}-Y_{F}\right)\sin\psi_{L}\\ d_{y}=\left(X_{L}-X_{F}\right)\sin\psi_{L}-\left(Y_{L}-Y_{F}\right)\cos\psi_{L} \end{array}$$
where the X and Y coordinates of the desired distance are $d_{x}$ and $d_{y}$, respectively. For the purposes of this study, an SMC controller is used to maintain formation topology in the face of external disturbances and environmental uncertainties. The *x* and *y* control errors should satisfy the following constraints:(17)$$\begin{array}{c} \lim_{t\to \infty }\|e_{x}\left\|=\right\|d_{x}^{d}-d_{x}\|=0\\ \lim_{t\to \infty }\|e_{y}\left\|=\right\|d_{y}^{d}-d_{y}\|=0 \end{array}$$
where $d_{x}^{d}$ and $d_{y}^{d}$ are the desired distance in the X and Y coordinates, respectively.

For this purpose, first-order SMC is introduced. The time varying surface S(t) is selected by s(e;t)=0, with:(18)$$\begin{array}{c} s\left(e,t\right)={\left(\frac{d}{dy}+\lambda \right)}^{n-1}e \end{array}$$

Then, the derivative function is given by:(19)$$\begin{array}{c} \dot{s}=\ddot{e}+\lambda \dot{e}\\ \frac{1}{2}\frac{d}{dy}s^{2}\leq -\eta |s| \end{array}$$

Finally, the formation topology is controlled for each agent as follows:(20)$$\begin{array}{c} {\ddot{X}}_{Fi}={\ddot{X}}_{L}+\lambda_{x}\left({\dot{X}}_{L}-{\dot{X}}_{Fi}\right)\\ {\ddot{Y}}_{Fi}={\ddot{Y}}_{L}+\lambda_{y}\left({\dot{Y}}_{L}-{\dot{Y}}_{Fi}\right) \end{array}$$

Once the position control problem is solved, a transformation to attitude control can be obtained. This means a minimum sharing data between agents:(21)$$\begin{array}{c} \theta_{Fi}=\theta_{L}+\lambda_{\theta }\left({\dot{\theta }}_{L}-{\dot{\theta }}_{Fi}\right)\\ \phi_{Fi}=\phi_{L}+\lambda_{\phi }\left({\dot{\phi }}_{L}-{\dot{\phi }}_{Fi}\right) \end{array}$$
with $\lambda_{\theta }>0$ and $\lambda_{\phi }>0$ being selective gains for the formation control.

### 4.3. Formation Transformation

This section describes the scenario when one of the swarm agents unexpectedly fails. In such situations, the agent performs an emergency landing to avert crashes, and the other agents must switch their topologies. Additionally, when the failed agent is a leader, a leader election procedure is required. This is carried out as outlined in Algorithm 1.
**Algorithm 1** Leader Election*N*: Number of agents, *A*: list of *N* agents where each one has an ID from 1 to *N*,  T={T(N), T(N−1), …,T(1)}: list of *N* topology according to the agent’s number, P(T): list of *N* paths according to each topology.
  1:idL=1  2:$L=A_{idL}$  3:Y=f(Batt,Health,Mission)  4:**while** true **do**  5:    $T_{ex}=T\left(N\right)$  6:    $P_{ex}=P\left(T_{ex}\right)$  7:    $L=[T_{ex},T,P_{ex},P,A,idL]$  8:    ALLReady=0  9:    **for** i←1
*N* 1 **do**10:        **if** i≠idL **then**11:           send(i,ready=false) by *L*12:           wait(i,ready(i))=true by *L*13:           ALLReady=ALLReady+114:        **end if**15:    **end for**16:    **if** AllReady=N−1 **then**17:        **for** i←1
*N* 1 **do**18:           **if** i≠idL **then**19:               $send(P_{ex}\left(i\right),i)$ by *L*20:           **end if**21:        **end for**22:        **if** Failure=idL **then**23:           IdY=024:           **for** j←1
*N* 1 **do**25:               **if** i≠idL **then**26:                   $State_{i}=election$27:                   $idY=max(Y\left(A_{i}\right),IdY)$28:               **end if**29:           **end for**30:           $L=A_{idY}$31:           idL=idY32:        **end if**33:        $A.remouveElement(A_{Failure})$34:        N=N−135:    **end if**36:**end while**


First of all, a set of *N* UAVs are identified from 1 to *N*. The desired topology (*T*) depends on the UAVs number. Each topology has its related paths (P(T)). The head of the UAV list ($A_{1}$) is considered a leader *L*. The leader is supposed to have the knowledge about: the UAVs IDs, the desired topology ($T_{ex}$), path ($P_{ex}$), and all the possible paths and topologies (*P* and *T*). The performance function (*Y*) depends on the battery remaining time, the UAV’s health (i.e., no failure presented), and the percentage of mission executing.

The leader sends a ready message to all UAVs before the mission to determine whether or not they are prepared (line 9). Once all UAVs are operational (line 15), the leader follows the reference path and maintains formation topology (line 18).

A test loop is required during the mission to test the state of each drone. If one of the UAVs fails (line 21), an emergency landing occurs to avoid a crash. Depending on the failing agent, one of two scenarios can occur. If the failing agent is a leader, the election procedure begins (line 25), and the agent with the best performance (*Y*) is chosen as the leader (line 26–31). If the failing agent is a follower, it will be deleted from the list of UAVs (line 32). Depending on the number of remaining UAVs, the intended topology will be switched in both scenarios (return to line 3).

## 5. Simulation Results

The simulation results for quadrotor creation using Matlab/Simulink are shown in this section. Many scenarios have been run based on formation control and the many limitations that can occur during a mission. All of the following scenarios make use of four quadrotor UAVs. The communication lines between all of the UAVs are expected to be secure. The goal is for all of the UAVs to be able to switch formations at any time. A summary of the simulated cases is provided below:**Scenario 1:** In this scenario, UAVs are deployed as fixed-position stationary nodes that serve as communication relays. Four UAVs under rectangular topology take off from various points and fly to a predetermined altitude. The goal is to keep the optimal fixed position for maximum network connectivity.**Scenario 2:** The mission to be executed during this scenario is that every UAV acts as an aerial MEC server. Nonetheless, one of the swarm agents has an unanticipated engine failure. The remaining UAVs must address this situation. Leader selection and topology switching are implemented.**Scenario 3:** The third scenario simulates the UAVs as mobile nodes in a Cellular-Connected MEC. Each UAV is designed to follow a desired path with different hovering positions to serve for IoT devices. The Swarm must travel to its intended destination while avoiding external obstacles and agents collisions.

### 5.1. Scenario 1: Relayed MEC

This scenario aims to test the ability of the designed framework to deploy the UAVs in a Relayed MEC. For this case, no leader is designed, and all the UAVs have knowledge about the desired path to follow. The UAVs from 1 to 4 start from the initial positions $F_{1}\left(0\right)={\left[10;-10;0\right]}^{T}$, $F_{2}\left(0\right)={\left[-10;-10;0\right]}^{T}$, $F_{3}\left(0\right)={\left[-10;10;0\right]}^{T}$ and $F_{4}\left(0\right)={\left[10;-10;0\right]}^{T}$, respectively, and meet each other at an altitude z=10m, while keeping a rectangular separation of 10 m. An extra edge controller is added to assure the convergence over the meeting points.

Figure 6 illustrates the success of UAVs in relaying MUs to the MEC server, as all quadrotors were able to reach the intended offloading spots and cover the entire proposed region. The formation errors between UAVs 1–2 and 3–4 converge to zero in less than six seconds. Due to the synchronization of the altitude control for all swarm agents, the error in altitude is kept at zero throughout the course of the mission. Such position hold accuracy, while preserving separation distances, would assist MUs in offloading their computationally intensive tasks and preventing connection overlap.

### 5.2. Scenario 2: Assisted MEC

In this scenario, the swarm is supposed to start from the following positions: $F_{1}\left(0\right)={\left[50;0;0\right]}^{T}$, $F_{2}\left(0\right)={\left[40;-10;0\right]}^{T},\;F_{3}\left(0\right)={\left[60;-10;0\right]}^{T}$, and $F_{4}\left(0\right)={\left[50;-20;0\right]}^{T}$. Every UAV is denoted as an MEC server to mobile users among its trajectory. For the mission, the leader is to achieve the desired point $P_{f}\left(x_{0},y_{0},z_{0}\right)={\left[50;100;10\right]}^{T}$. The swarm is supposed to hold the diamond topology with a separation distance of 10 m between the agents. At t=25 s, an engine failure occurs to the leader. According to Algorithm 1, the leader lands to avoid the crash, a new leader is then elected, and the formation is switched from diamond to triangular. Depending on the elected leader, three possible cases can be carried out:

#### 5.2.1. Case 1

Let us suppose that *Follower* 1 is elected as a new leader. As depicted in Figure 7, the swarm was able to switch its topology from diamond to triangular immediately following the leader’s emergency landing. Figure 8 demonstrates that the position transfer between the new leader and *Follower* 2 took only 5 s, but the separation distance between the new leader and *Follower* 3 (10 m) was strictly observed during the switching process. After t = 30 s, the new triangular topology was totally established.

#### 5.2.2. Case 2

In this case, *Follower* 2 is supposed to be elected as a new leader. Figure 9 and Figure 10 illustrate the simulation case. The obtained results are similar to those obtained in the first case. The triangular topology was achieved in t = 30 s, and the position’s switching between the new leader and *Follower* 3 was made in 5 s. The separation distance between the new leader and *Follower* 1 was also maintained during the switching operation (Figure 10).

#### 5.2.3. Case 3

For this last case, *Follower* 3 is supposed to be elected as the new leader. As shown in Figure 11, *Follower* 3 is positioned behind the leader, thus, there is no need to switch positions. Figure 12 demonstrates that the separation distance was maintained with *Follower* 3 and *Follower* 1 with high accuracy.

### 5.3. Scenario 3: Cellular-Connected MEC

The UAVs in this scenario are considered as mobile nodes in a cellular-connected MEC server. The four UAVs are supposed to start from the following positions: $F_{1}\left(0\right)={\left[50;65\right]}^{T}$, $F_{2}\left(0\right)={\left[60;75\right]}^{T}$, $F_{3}\left(0\right)={\left[50;85\right]}^{T}$, and $F_{4}\left(0\right)={\left[40;75\right]}^{T}$. The leader’s objective is to get to the target spot: $P_{f}\left(x_{0},y_{0}\right)={\left[50;15\right]}^{T}$ and avoid the obstacle (R = 10 m) situated at $O_{1}\left(x_{0},y_{0}\right)={\left[50;50\right]}^{T}$. The agents are separated by 10 m. The diamond configuration is expected to be held by the swarm. Otherwise, it is possible to split the topology, avoid the obstacles, and go back to the initial formation.

In order to conserve energy, the swarm was able to circumvent the circular barrier by forming a distributed configuration (Figure 13). The swarm was split into two teams, each with two leaders: one created by the original leader and follower 1, and the other by follower 2 (new leader) and follower 3 (Figure 14). The formation error of teams 1 and 2 is shown below. It can be seen that the separation distance (10 m) between the two UAVs was respected with excellent precision in both the x- and y-directions. This demonstrates the formation controller’s excellent performance.

Table 1 summarizes the results of the previous scenarios. As previously demonstrated, the UAV swarm proved capable of enabling diverse MEC topologies and accommodating all possible constraints. While tracking the created path with the least possible error/delay, the convergence time was very reasonable. The architecture is also ideal for long-range missions due to the energy consumption strategy.

### 5.4. Comparative Study

To show the usefulness of the proposed framework, a comparison with state-of-the art methods was conducted. The comparison of the architectures’ key features is presented in Table 2, from which we can deduce that our suggested framework constitutes a comprehensive effort in comparison to other current works. First, the distributed architecture solves the shortcomings of the centralized architecture since leader and follower interaction is taken into account. In addition, the algorithms utilized for path planning and formation control are user-friendly, suitable for a large number of robots, and can be implemented within an embedded system. Our framework is the only one that has strengthened safety precautions by including a leader election process.

In addition, the performance of formation control results may be found in Table 3. We can argue that the employment of attitude-based control based on SMC controller has significantly enhanced the L-F configuration’s precision, speed, and stability.

## 6. Conclusions

The issue of multi-UAV-enabled MEC architectures in a distributed formation was investigated in this study. In the case of centralized UAV formation, the leader is the sole agent who is fully aware of the mission’s details. As a result, every probable failure has an impact on mission execution. To overcome this issue, a new framework of quadrotors with a distributed L-F configuration was presented. The framework is divided into several stages, each of which takes into account the relationship between the leader and the followers. Throughout the formation control stage, a consensus-based attitude control was used to maintain the formation topology with minimal data sharing. An SMC controller was used to monitor the optimal generated path with the minimum possible error/delay. During the trajectory generation phase, a collaborative obstacle avoidance technique was developed for safety and to accommodate the multiple environmental constraints. The swarm’s agents were able to switch topology, avoid obstacles, and return to the desired formation without colliding. Furthermore, in response to an agent’s unexpected death, the agents were able to elect new leadership. The leader election phase was designed to indicate topological changes caused by external obstacles. In comparison to the vast majority of comparable suggested works, all produced results were deemed satisfactory. Although this study focused on the use of UAVs under various MEC architectures, the technical computing approach can be generalized and applied to many other IoT specifications and access control strategies, such as networking and data gathering approaches, educational platforms, healthcare systems, transportation services, and many other real-world applications. Therefore, incorporating the proposed architecture into an MEC-based real-world application would be an intriguing endeavor.

## Figures and Tables

**Figure 1 sensors-22-07243-f001:**
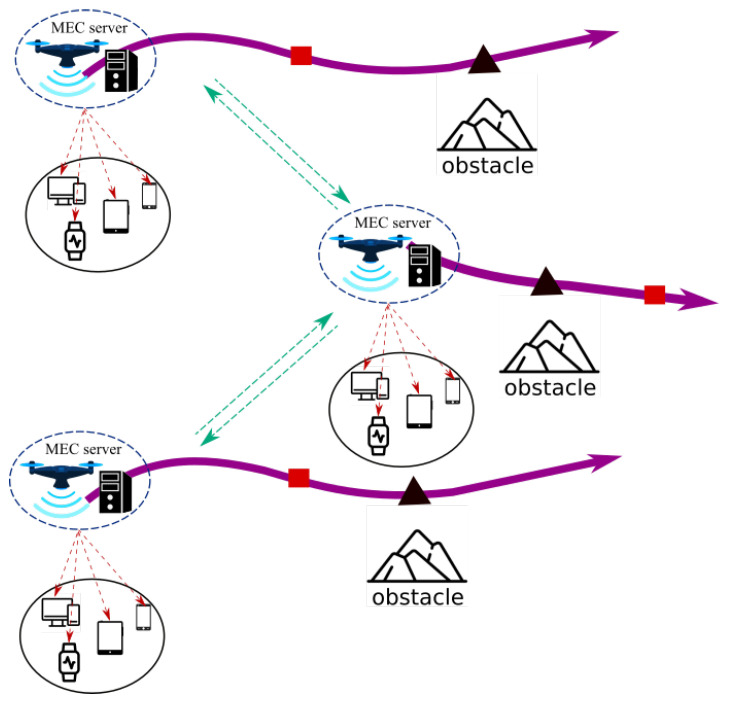
Assisted MEC architecture.

**Figure 2 sensors-22-07243-f002:**
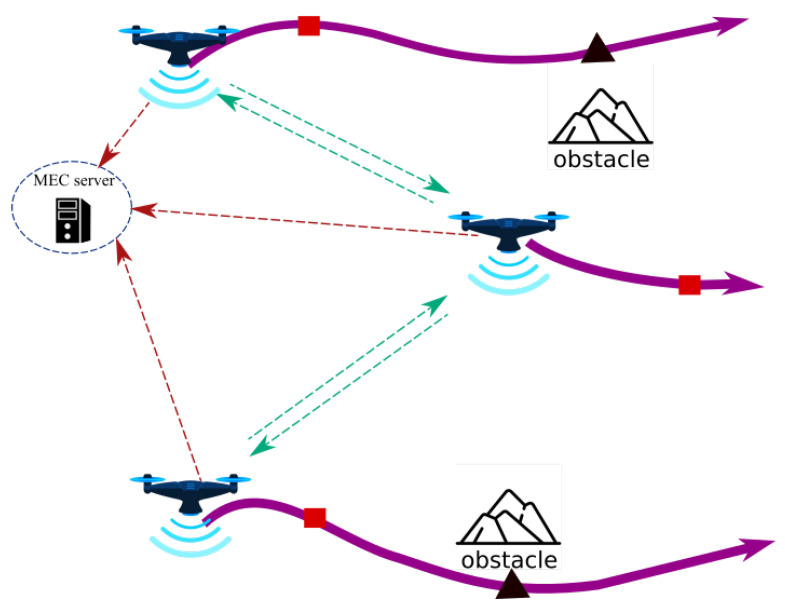
Cellular-Connected MEC architecture.

**Figure 3 sensors-22-07243-f003:**
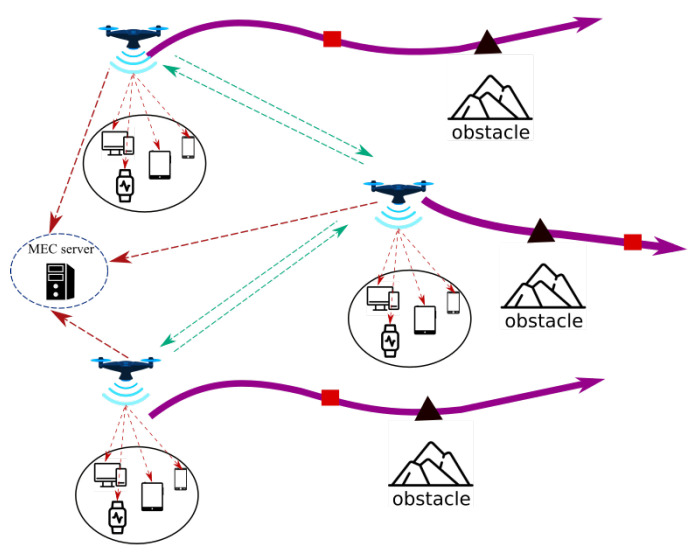
Relayed MEC architecture.

**Figure 4 sensors-22-07243-f004:**
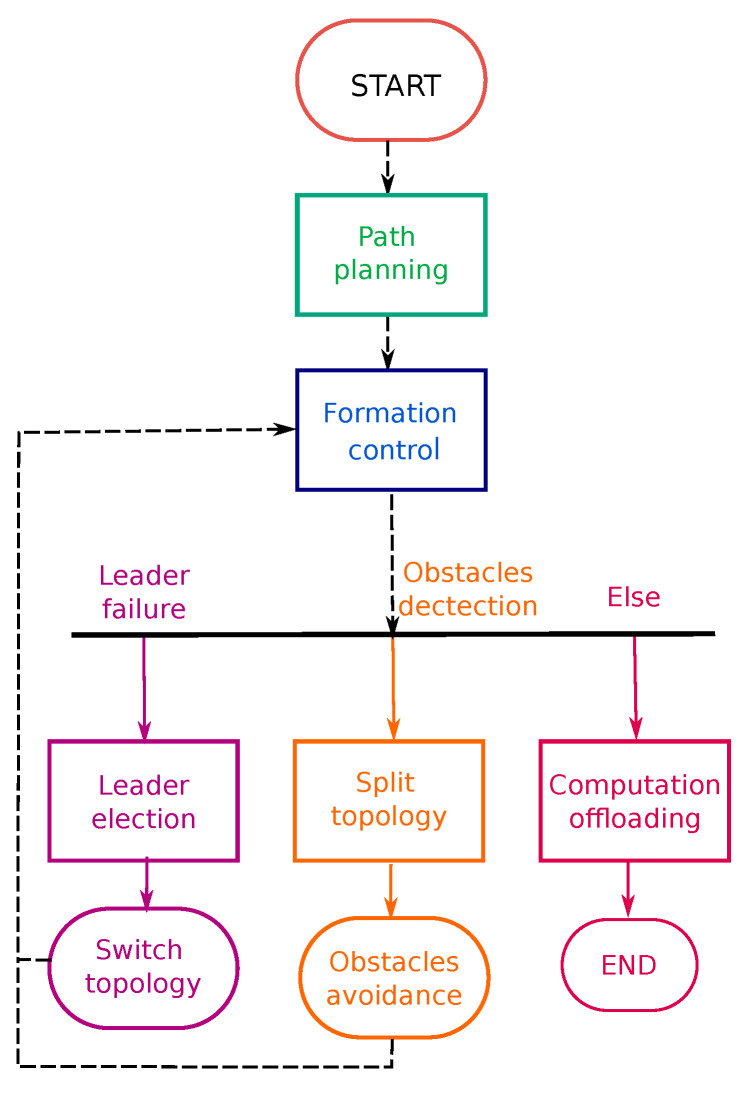
Proposed framework for UAVs swarm monitoring under distributed L-F formations.

**Figure 5 sensors-22-07243-f005:**
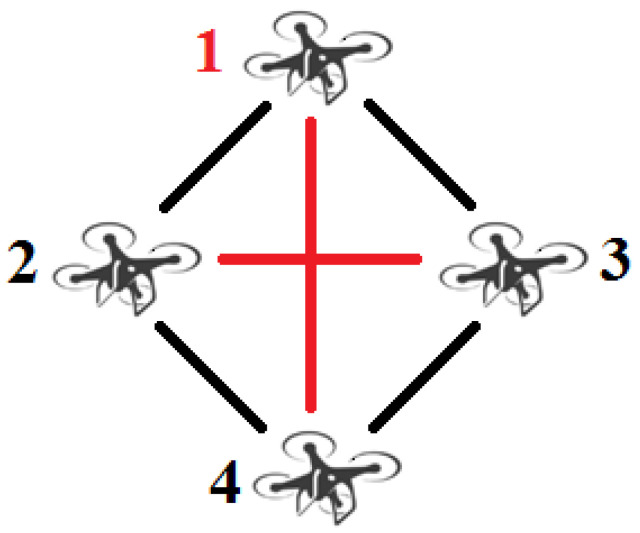
Leader–follower formation topology.

**Figure 6 sensors-22-07243-f006:**
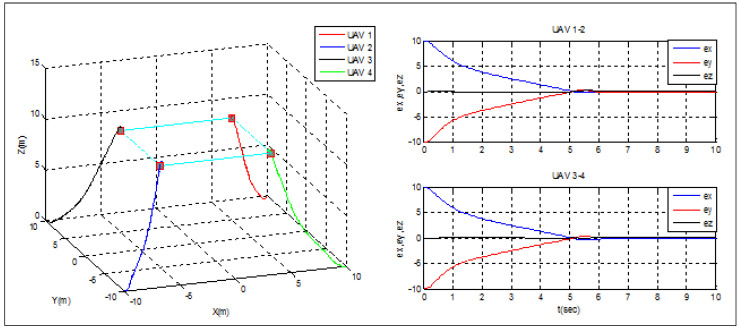
Relayed MEC under distributed formation.

**Figure 7 sensors-22-07243-f007:**
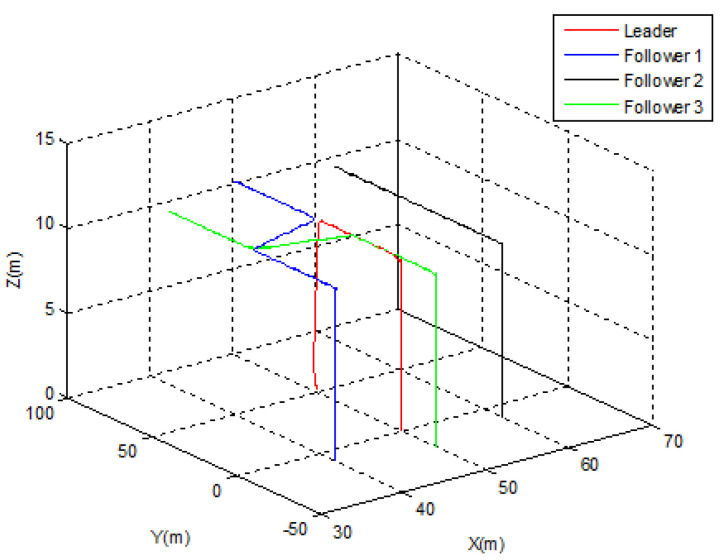
Leader election Case 1.

**Figure 8 sensors-22-07243-f008:**
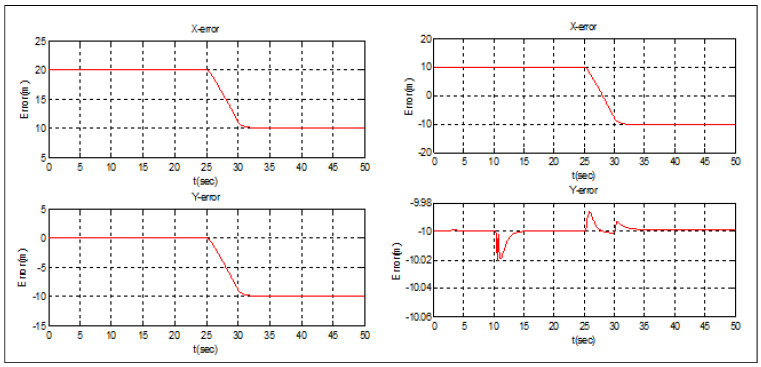
Leader election Case 1—tracking errors.

**Figure 9 sensors-22-07243-f009:**
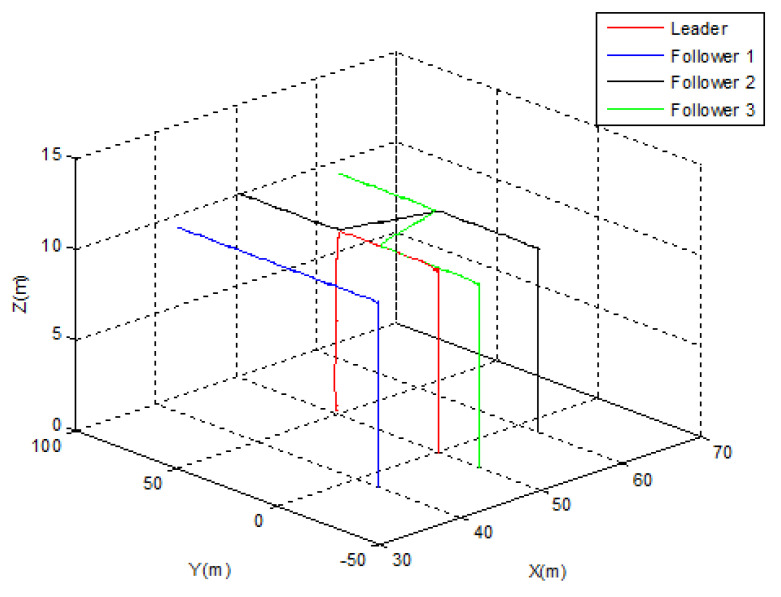
Leader election Case 2.

**Figure 10 sensors-22-07243-f010:**
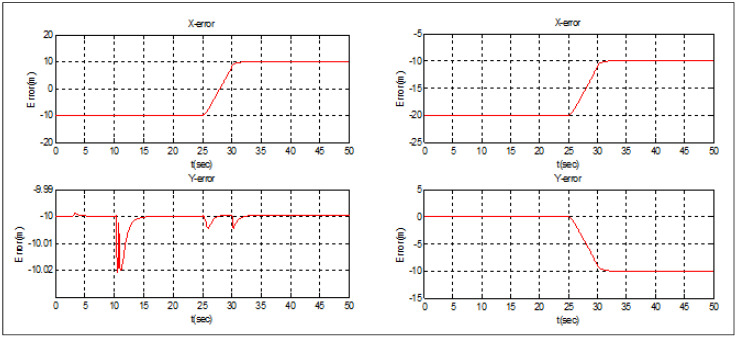
Leader election Case 2—tracking errors.

**Figure 11 sensors-22-07243-f011:**
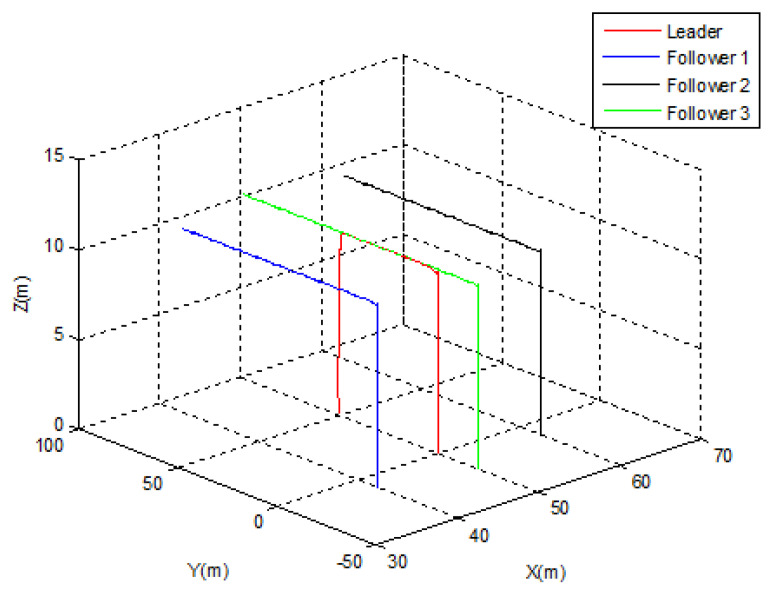
Leader election Case 3.

**Figure 12 sensors-22-07243-f012:**
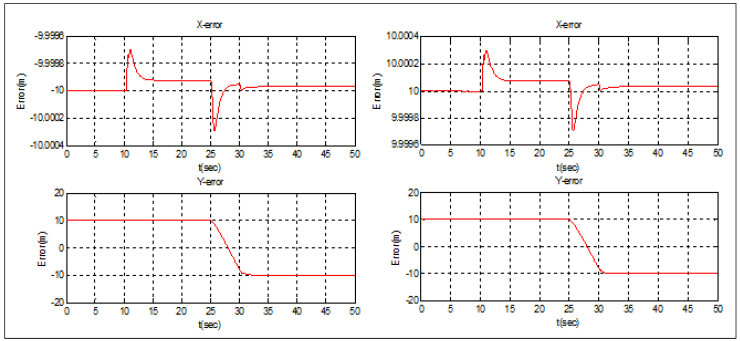
Leader election Case 3—tracking errors.

**Figure 13 sensors-22-07243-f013:**
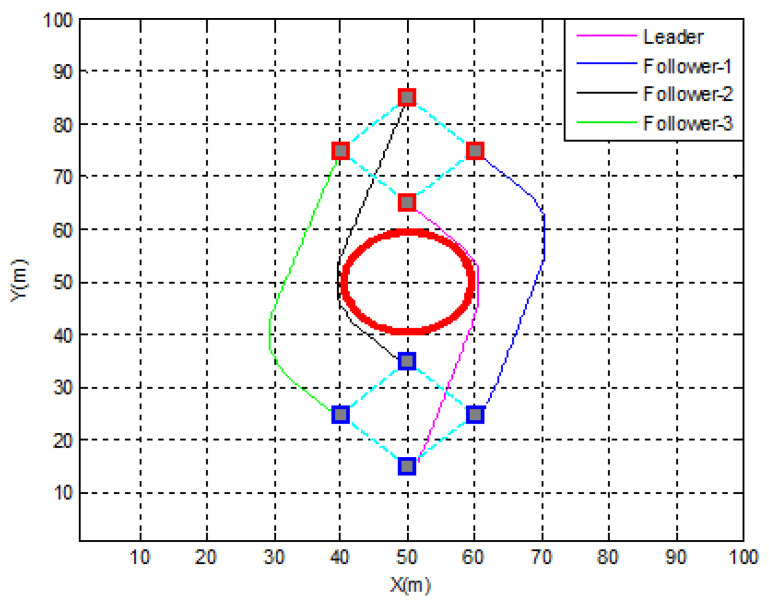
Obstacles avoidance.

**Figure 14 sensors-22-07243-f014:**
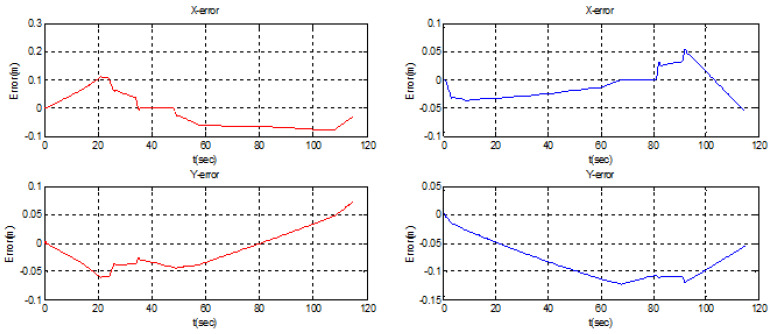
Obstacles avoidance tracking error.

**Table 1 sensors-22-07243-t001:** Performance results.

MEC Architecture	Relayed	Assisted	Cellular-Connected
**Convergence time (s)**	10	50	120
**Traveled distance (m)**	10	70	80
**Total energy consumption (%)**	5	15	20

**Table 2 sensors-22-07243-t002:** Article architecture comparison.

Author		Wu et al. (2020)	Wen et al. (2019)	Tran et al. (2021)	Proposed Framework
**Formation**		Centralized	Decentralized	Distributed	Distributed
**Vehicle**	**Type**	UAV	UGV	UAV/UGV	UAV
	**Number**	8	4	3	4
**Path planning**		PSO	APF	NI	CO
**Formation control**	**Strategy**	Position consensus	Position consensus	Velocity consensus	Attitude consensus
	**Controller**	MPC	Robuste H *∞*	NI	SMC
**Safety precautions**		N/C	Switching	Switching	Switching/Leader election

**Table 3 sensors-22-07243-t003:** Formation control performance comparison.

Author		Wu et al. (2020)	Wen et al. (2019)	Tran et al. (2021)	Proposed Framework
**Rise time (s)**		5	3	5	4
**Over shoot %**		0	0	5	0
**Setting time (s)**		10	5	10	5
**Switching time (s)**		N/C	4	5	1
**Tracking error (m)**	**x**	N/A	0.3	0.1	0.05
	**y**	N/A	0.3	0.1	0.05

## Data Availability

No data available.

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
