# Peer review of "Path Planning and Formation Control for UAV-Enabled Mobile Edge Computing Network"

_sensors, 2022, doi:10.3390/s22197243_

Round 1

Reviewer 1 Report

Dear authors,
I have reviewed the paper entitled "Path planning and Formation Control for UAV-Enabled Mobile Edge Computing Network". 

The implementation seems to the reviewer a good software engineering effort and the various choices of components of the framework used are instructive to a novice reader. However, for publication as a research article, the novel research contribution leaves much to be desired.

Why the proposed algorithm is it not compared with other SoA methods?

Author Response

The implementation seems to the reviewer a good software engineering effort and the various choices of components of the framework used are instructive to a novice reader. However, for publication as a research article, the novel research contribution leaves much to be desired.

Why the proposed algorithm is it not compared with other SoA methods?

We would like to thank the reviewer for taking time to assess our manuscript. As described in our work, we deal with the problem of path planning and formation control when using a swarm of drones. The problem has been tackled from artificial intelligence and automation point of view rather from software engineering point of view.  The comparative study has been conducted using most related work. It may be a good attempt to compare to SoA methods as suggested by the reviewer but this is out of scope of our work.

Reviewer 2 Report

In this paper, the formation task of UAVs was studied by using MEC, and the Leader-Follower method was adopted to construct the formation, and the problem of relying too much on the leader was innovated. This paper has some innovations, but also has some shortcomings. The comments for this paper are shown in the followings:

1. The specific use process of MEC is not reflected in the simulation. There should be some verification of the computational power of the system. The layout of the figure does not correspond to the text.

2. In the introduction of graph theory in Section 2, it is more appropriate to describe the L-F method by a directed graph model.

3. The optimized design in Section 3 is reasonable, but the constraint representation is questionable. In Equation 7,  represents the radius of the obstacle avoidance area, should it be greater than this radius?  should be  in Equation 6.

4. The paper structure needs to be improved. The suggestions are as follows:

(1) The introduction of research background in the Abstract is a little too much.

(2) The Leader Election in Section 5 should be used as the formation transformation part of formation control.

5. The expression needs to be improved. The suggestions are as follows:

(1) "Equation" in the text does not have an uppercase "E".

(2) Superscripts G, A, etc., should be positive. The transpose sign should be positive.

(3) The interpretation of the formula 'where' should not be indented.

Author Response

In this paper, the formation task of UAVs was studied by using MEC, and the Leader-Follower method was adopted to construct the formation, and the problem of relying too much on the leader was innovated. This paper has some innovations, but also has some shortcomings. The comments for this paper are shown in the followings:

We would like to thank the reviewer for taking time to review our manuscript and for his insightful comments

  1. The specific use process of MEC is not reflected in the simulation. There should be some verification of the computational power of the system. The layout of the figure does not correspond to the text.

The specific use process of MEC is not reflected in the simulation: Indeed, the focus of our work was on the challenges that must be addressed whenever a swarm of drones must be deployed, regardless of the task at hand. These challenges have more to do with swarm design, member coordination, and formation control. This explains our simulation process. The three situations analyzed in our simulation studies show how our suggested framework might be implemented in a MEC setting. Reflecting more on the MEC usage process necessitates additional research and simulations, which could result in a new body of work and be a potential future work as we suggested at the end of the conclusion (page 17 line 424).

" Therefore, incorporating the proposed architecture into a MEC-based real-world application would be an intriguing endeavor."

There should be some verification of the computational power of the system. We thank the reviewer for this important point. Verifications over the computational power of the system and other performance measures of the proposed scenarios are provided in Table.1 Page 15.

The layout of the figure does not correspond to the text. Modifications were made to the simulation results section as needed to guarantee text-figure correspondence which you can see in the highlighted texts throughout the document.

  1. In the introduction of graph theory in Section 2, it is more appropriate to describe the L-F method by a directed graph model.

The use of a directed graph model forces nodes to communicate in a specified direction which is not suitable for the L-F method we want to develop. Agents can move in any direction with regard to the used topology. To accommodate this situation an undirected graph is more suitable.

  1. The optimized design in Section 3 is reasonable, but the constraint representation is questionable. In Equation 7, represents the radius of the obstacle avoidance area, should it be greater than this radius?  should be in Equation 6.

We sincerely appreciate the reviewer's insightful comment. It is, in fact, a careless mistake. To avoid collision, the radius should be bigger than the radius of the obstacle avoidance area. This has been corrected. Equation 7 on page 8 line 244.

  1. The paper structure needs to be improved. The suggestions are as follows:

(1) The introduction of research background in the Abstract is a little too much.

The abstract has been updated accordingly. The part devoted to background has been reduced as follows:

Before:

Recent developments in unmanned aerial vehicles (UAVs) or drone technology have led to the introduction of a wide variety of innovative applications and services especially in the Mobile  Edge Computing (MEC) field. UAVs swarms are suggested as a promising solution to cope with  the issues that may arise when connecting Internet of Things (IoT) applications to a fog platform.  A swarm of drones has many advantages over a single drone, including increased reliability and  decreased operation time. One important aspect in the design of a UAVs swarm is the coordination  between swarm agents in complex environments with obstacles which is a recent research challenge  being investigated by the scientific community. Due to its simplicity and effectiveness, centralized  leader-followers formation is one of the most prevalent architectural designs in the literature. In the  event of a failed leader, however, the entire mission is canceled.

After:

Recent developments in unmanned aerial vehicles (UAVs) have led to the introduction of a wide variety of innovative applications especially in the Mobile Edge Computing (MEC) field. UAVs swarms are suggested as a promising solution to cope with the issues that may arise when connecting Internet of Things (IoT) applications to a fog platform. We are interested in a crucial aspect of designing a swarm of UAVs in this work, which is the coordination of swarm agents in complicated and unknown environments. Centralized leader-followers formation is one of the most prevalent architectural designs in the literature. In the event of a failed leader, however, the entire mission is canceled.

 (2) The Leader Election in Section 5 should be used as the formation transformation part of formation control.

The leader election section is included under the formation control as formation transformation. See Section 4.3, Line 287 on page 10

  1. The expression needs to be improved. The suggestions are as follows:

(1) "Equation" in the text does not have an uppercase "E".

All ‘equations’ with lowercases are corrected. See Page 7, line 209,212 and Page 9, line 257.

(2) Superscripts G, A, etc., should be positive. The transpose sign should be positive.

Interaction matrix are supposed to be positive as their transpose. See Page 6, Line 196,205.

(3) The interpretation of the formula 'where' should not be indented.

Corrections attributed to indented ‘Where’ formula. See Page 7, Line 209 ; Page 8, Line 246 and Page 9, Line 263.

Round 2

Reviewer 1 Report

The authros have revised the manuscript accordingly. I suggest this paper can be accepted now.